# Local and Systemic Endothelial Damage in Patients with CEAP C2 Chronic Venous Insufficiency: Role of Mesoglycan

**DOI:** 10.3390/ijms26094046

**Published:** 2025-04-24

**Authors:** Angelo Santoliquido, Claudia Carnuccio, Luca Santoro, Angela Di Giorgio, Alessia D’Alessandro, Francesca Romana Ponziani, Flavia Angelini, Marcello Izzo, Antonio Nesci

**Affiliations:** 1Angiology and Noninvasive Vascular Diagnostics Unit, Department of Cardiovascular Sciences, Fondazione Policlinico Universitario Agostino Gemelli IRCCS, 00168 Rome, Italy; angelo.santoliquido@policlinicogemelli.it (A.S.); luca.santoro@policlinicogemelli.it (L.S.); angela.digiorgio@policlinicogemelli.it (A.D.G.); alessia.dalessandro@policlinicogemelli.it (A.D.); antonio.nesci@policlinicogemelli.it (A.N.); 2Department of Translational Medicine and Surgery, Catholic University of the Sacred Heart, 00168 Rome, Italy; francescaromana.ponziani@policlinicogemelli.it (F.R.P.); flavia.angelini@policlinicogemelli.it (F.A.); 3Digestive Disease Center (CEMAD), Fondazione Policlinico Universitario Agostino Gemelli IRCCS, 00168 Rome, Italy; 4Compression Therapy Study Group (CTG), Math Tech Med Medicine & Bioscience, Research Center, University of Ferrara, 44121 Ferrara, Italy

**Keywords:** chronic venous disease, systemic inflammation, glycocalyx, syndecans, mesoglycan

## Abstract

Chronic venous disease (CVD) involves complex pathophysiological mechanisms, particularly an imbalance between matrix metalloproteinases (MMPs) and their tissue inhibitors (TIMPs), contributing to venous remodeling and varicosities. Elevated MMP-2 and MMP-9 levels are commonly found in tissues affected by venous ulcers. Inflammation plays a central role in CVD, with higher levels of pro-inflammatory markers present in varicose veins compared to healthy ones. Syndecans, key components of the endothelial glycocalyx, are involved in inflammatory responses. Alterations in the glycocalyx structure are associated with vascular damage in both venous and arterial diseases. This study aimed to investigate inflammatory changes in CVD patients, focusing on glycocalyx damage and the therapeutic role of mesoglycan, a glycosaminoglycan-based drug. A prospective, monocentric study included 23 patients with C2 clinical–etiological–anatomical–pathological (CEAP) CVD. Serum samples were collected before and after mesoglycan treatment. Results showed significantly elevated levels of VCAM-1, MMP-2, MMP-9, SDC-1, IL-6, and IL-8 in blood from varicose veins versus the systemic circulation. Patients received 50 mg of mesoglycan orally every 12 h for 90 days. After treatment, a notable reduction in inflammatory markers was observed. These results support the hypothesis that mesoglycan may alleviate both local and systemic inflammation, providing insights into new therapeutic strategies for CVD management.

## 1. Introduction

Chronic venous disease (CVD) is a pathological condition that refers to structural and functional alterations of the venous system. Temporal evolution of CVD in the lower legs is a chronic disease that can manifest with various clinical patterns, ranging from telangiectasias and varicose veins to the development of marked skin changes and venous ulcers. CVD is characterized by a high prevalence in the general world population, estimated at around 73% in women and 53% in men, and has a high global economic impact [1]. From the mildest manifestations to the most severe forms, CVD leads to a significant deterioration in the quality of life of the affected subjects.

The American Venous Forum Consensus Report in 1994 presented the clinical–etiological–anatomical–pathological (CEAP) classification, which was updated in 2004 and recently in 2020. This classification shows the standardized clinical signs for the correct identification of CVD [2]. In the CEAP classification, class C2 indicates the presence of varicose veins and is considered the first clinical stage of chronic venous insufficiency (CVI). Varicose veins are dilated, tortuous superficial veins that reflect underlying venous dysfunction. Class C2 can also be subclassified as C2s (symptomatic) or C2a (asymptomatic) based on the presence or absence of typical symptoms of chronic venous disease, such as pain, heaviness, burning, itching, cramps, or restless legs.

Clinical manifestations of CVD recognize a unique physiopathology represented by venous stasis and venous hypertension, which can be caused by shear stress, venous reflux, and valvular incompetence [3]. Venous hypertension causes endothelial damage because it is responsible for a vicious circle that involves the cascade activation of endothelial cells (ECs) and inflammatory cells. The dysfunctional and activated ECs secrete a series of mediators and express a series of adhesion molecules, triggering a cascading mechanism of activation of inflammatory cells that support the damage to the endothelium and surrounding tissues.

In this inflammatory microenvironment, high levels of multiple cytokines, inflammatory molecules, selectins, and vasoactive factors are recognized, including monocyte chemoattractant protein 1 (MCP-1), tumor necrosis factor-α (TNF-α), intercellular adhesion molecule-1 (ICAM-1), vascular cell adhesion molecule-1 (VCAM-1), vascular endothelial growth factor (VEGF), and interleukins (IL-6, IL-8, and IL-1β). Sustained inflammation alters the homeostasis of the venous circulation, which is manifested in structural and functional modifications of vessel walls and the venous valve system [4]. In this context, the endothelial glycocalyx plays a crucial role. Activation of the inflammatory cascade and proteolytic enzymes, such as matrix metalloproteinases (MMPs) and seroproteinase, released by endothelial cells and leukocytes, and the decrease in tissue MMP inhibitors (TIMPs), cause the complete loss of the glycocalyx or partial glycocalyx shedding [4]. Therefore, a damaged endothelium allows activated leukocytes to migrate into the extracellular matrix and to release fibroblast growth factor-β (FGF-β) and cytokine transforming growth factor β-1 (TGF-β1) [5]. MMPs promote cellular damage and degrade the extracellular matrix, causing valvular dysfunction and damage to the vascular walls. In fact, studies show that MMP-2 and MMP-9 levels are particularly high in the tissues of subjects with venous ulcers in the active phase [6], and that levels are reduced significantly with the healing of the ulcer [7].

A variety of inflammatory and other biomarkers have been explored in the literature as potential indicators of clinical disease severity and predictors of progression to more advanced stages, yielding promising results [8,9].

Recent studies have identified that structural components of the endothelial glycocalyx can be considered markers of endothelial phlogosis [10]. The endothelial glycocalyx is a grass-like layer of glycosaminoglycans (GAGs) and structural proteins that form an extracellular barrier around the ECs. The glycocalyx acts as an organ that transduces extracellular stimuli and signals, regulates vascular permeability and tone, controls platelet and leucocyte adhesion, balances coagulation and fibrinolysis, and protects the endothelium from ROS and proteolytic enzymes [11]. It is possible to distinguish an apical glycocalyx from a basal glycocalyx, as the composition and function of these two surfaces differ. Specifically, the apical portion is mainly composed of glycoproteins and proteoglycans. Syndecan-1 (SDC-1), a key component of the apical glycocalyx, transmits mechanical and inflammatory stress signals to endothelial cells, while syndecan-4 (SDC-4), predominant in the basal glycocalyx, interacts with adhesion molecules and integrins to enable cytoskeletal polarization in response to blood flow. These two surfaces are closely interconnected. It has been shown that these isoforms of the syndecans play important regulatory roles in wound healing [12,13], low-grade inflammation [4], angiogenesis [14], and oxidative stress in patients with trauma or sepsis [15,16].

Changes in blood flow and altered shear stress, which occur in atherosclerosis, are implicated in the alteration of the glycocalyx [17]. It is interesting to note that some studies show that the glycocalyx alterations found in atherosclerosis are not identified for each type of plaque (i.e., large calcification and cholesterol crystal plaques), but they are more associated with unstable plaques [18].

Higher serum levels of IL-6, in the absence of acute inflammatory or infectious disease, are significantly associated with lower flow-mediated dilation (FMD) values in individuals considered healthy and at low cardiovascular risk [19]. This study supports the notion that certain blood markers can be detected early, long before routine diagnostic or biochemical assessments, allowing the identification of individuals who are statistically more likely to develop cardiovascular damage over time compared to those with normal or low IL-6 levels. FMD serves as an early instrumental marker, detectable long before abnormalities are evident through methods such as color Doppler ultrasounds [19].

In parallel with CVD, endothelial damage and dysfunction are early markers of arterial diseases. In fact, they can be detected well before the detection of macroscopic manifestations, such as for ultrasonographic imaging of an atherosclerotic plaque [20]. Changes in cytokine and endothelial marker levels can be found in arterial diseases. For example, MMPs may be used as markers of cardiovascular risk due to their proteolytic properties, which have a destabilizing effect on atherosclerotic plaque [21]. Recently, several authors highlighted a close connection between venous and arterial pathologies [22,23]. In Prochaska et al., CVD was independently associated with arterial hypertension, peripheral artery disease, and venous thromboembolism, with a higher prevalence of cardiovascular disease as the severity of the CEAP classification increases [24]. CVD has been found to be a strong predictor of all-cause mortality. Although a causative relationship between CVD and cardiovascular disease has not yet been discovered, the research of Prochaska et al. lays the groundwork for thinking about venous insufficiency as an additional risk factor for cardiovascular diseases, regardless of traditional cardiovascular risk factors.

The pharmacotherapy of CVD is used in all stages of the disease, both to counteract the physiopathological mechanisms that support tissue damage and to improve the severity of symptoms and quality of life and prevent the most unfortunate consequences of the disease. In this context, a wide range of venoactive drugs are recognized, comprising both products of natural origin and synthetic drugs. Many phlebotonics, such as diosmin, hesperidin, Ruscus, and others, have shown proven beneficial effects in CVD by interacting with key pathogenic mechanisms; however, the effect of any of these drugs on the cellular glycocalyx has not been evaluated [25,26]. Other drugs with proven effects on vascular wall remodeling mechanisms include statins due to their pleiotropic effects. Indeed, they have been shown both in vitro and in vivo to interfere with wall stress–mediated activator protein 1 (AP-1) activity in venous smooth muscle cells [27,28].

Given the importance of the glycocalyx in various physiological functions, its protection may represent a promising target in the treatment of chronic vascular diseases such as CVD. Animal-derived products refer to a series of mixtures of glycosaminoglycans that play a role in maintaining glycocalyx integrity by delivering precursors for its various constituents [29]. Two compounds are distinguished: sulodexide and mesoglycan. Sulodexide is a mixture of highly purified GAGs, extracted from the porcine intestinal mucosa, consisting of low-molecular-weight heparin (80%) and dermatan sulfate (20%) [30]. Mesoglycan is a natural preparation composed of a mixture of GAGs (47.5% heparan sulfate, 35.5% dermatan sulfate, 8.5% chondroitin sulfate, and 8.5% slow-mobility heparin) with a negative electric charge and extracted from the porcine intestinal mucosa. Mesoglycan is frequently used in common clinical practice for its recognized clinical benefits in CVD, as well as in post-thrombotic syndrome, and for the recognized effects of the disease-modifying agent [31]. In particular, mesoglycan has been extensively studied in patients with stage 6 CVD [32]. Mesoglycan exerts its pharmacological activity at the endothelial level, restoring the electronegativity of the damaged endothelium and the endothelial glycocalyx, thus restoring the integrity of the capillary membrane and maintaining extracellular fluid homeostasis [30].

Recent data show how the use of mesoglycan is effective in improving endothelial function in peripheral artery disease through a laboratory demonstration of a reduction in MMPs and other endothelial markers [33]. However, there is no evidence on its use within the early stages of CVD.

The objectives of our study are (1) to confirm the presence of inflammatory changes in patients with CVD, regarding the role of the SDCs in the assessment of glycocalyx damage both locally and systemically, and (2) to assess the effect of mesoglycan administration on the local and systemic inflammatory states.

## 2. Results

### 2.1. Materials, Population, and Treatment

During the study period, 187 consecutive patients who attended our center were observed, of whom 84 met the inclusion criteria; of these, 27.5% participated in this study and provided informed consent. Thus, this study was conducted on a group of 23 patients with CVD classified as CEAP C2. The study population consisted of eighteen (78.2%) women and five (21.7%) men, with a median age of 60 (IQR, 23–74) years and predominantly of Caucasian ethnicity (No. [%]; 22, [95.6%]).

The median height was 165 (IQR, 158–192) cm, and the median body weight was 68 (IQR, 52–102) kg. Of the enrolled patients, 19 (82.6%) reported having one or more first- or second-degree relatives with CVD. Among female patients, 13 (72.2%) reported a history of one or more pregnancies. A total of nine (39.1%) of patients had a BMI > 25 kg/m^2^, and fifteen (65.2%) were sedentary. Six (26%) patients were smokers (Table 1).

All patients presented varicose veins in the lower limbs, and 22 (95.6%) reported symptoms consistent with CVD. On the Doppler ultrasound (DUS), the following findings were recorded: incompetence of the great saphenous vein (GSV) in thirteen (56.5%) (reflux > 0.5 s), incompetence of the small saphenous vein (SSV) in seven (30.4%), and anterior accessory GSV incompetence in six (26%). In 17 (73.9%) patients, varicose collateral veins were documented in the region of the GSV proximal to the knee, and in 21 (91.3%) patients, collateral veins were present distal to the knee. Varicose collaterals were also found in eight (34.7%) patients in the region of the small saphenous vein. No patient showed clinical or ultrasound signs of DVT, SVT, or EP. The most common comorbidities reported were hypothyroidism (21.7%), followed by dyslipidemia (17.3%), a history of breast cancer (8.6%), endometriosis (4.3%), benign prostatic hyperplasia (4.3%), undetermined dermatitis (4.3%), and hypertension (4.3%). Ten (43.5%) patients had no comorbidities. The most used medications were levothyroxine (17.3%), estroprogestins (8.6%), nebivolol (4.3%), aspirin (4.3%), hydroxychloroquine (4.3%), and minoxidil (4.3%). Fifteen (65.2%) patients were not on any home medications. No patients had a history of anticoagulant therapy prior to or during the study. Eight (34.7%) patients wore elastic compression stockings.

Blood samples were collected at baseline (T0) and after treatment with mesoglycan (T1) from both the antecubital vein and the most distal varicose vein of the lower limb, with the patient in the standing position. No complications related to venous sampling occurred at either T0 or T1. A total of 65.2% of blood draws from varicose veins were from the left lower limb, and in 91.3% of cases, the sampling was from veins distal to the knee.

Twenty (86.9%) of the enrolled patients started oral mesoglycan therapy at 50 mg every 12 h for a median of 84 (IQR, 14–112) days. Three (13%) patients were non-compliant with the treatment. Two (8.6%) patients were excluded from this study due to a SARS-CoV-2 infection (4.3%) and an SVT episode (4.3%). No causal relationship was found between the onset of SVT and venous sampling. In three (13%) patients, adverse effects were reported, with the most common being dyspepsia. This led to a reduction in the mesoglycan dose to 50 mg every 24 h in two patients and early discontinuation of the treatment in one patient.

### 2.2. Serum Levels of Parameters at Time 0

Serum analysis of patients with varicose veins revealed higher levels of VCAM-1 (763.5 [685.7–849.1] ng/mL vs. 709.9 [627.4–826.0] ng/mL), MMP-2 (51.48 [46.94–88.39] ng/mL vs. 43.57 [37.23–79.19] ng/mL), MMP-9 (69.02 [39.89–96.74] ng/mL vs. 52.91 [36.87–94.26] ng/mL), SCD-1 (187.4 [167.2–235.7] ng/mL vs. 177.2 [161.7–203.2] ng/mL), IL-6 (3.89 [1.95–5.20] pg/mL vs. 2.57 [1.67–3.77] pg/mL), IL-8 (9.16 [6.86–14.76] pg/mL vs. 8.10 [4.91–17.96] pg/mL), ICAM-1 (23.72 [21.64–29.65] ng/mL vs. 23.18 [18.43–28.55] ng/mL), TGF-β (31.29 [20.55–46.15] ng/mL vs. 29.98 [21.98–44.32] ng/mL), and SDC-4 (56.75 [46.04–71.71] ng/mL vs. 51.74 [38.27–62.31] ng/mL) compared to serum samples from the systemic circulation. In varicose veins, TIMP-2 levels were lower (127.29 [98.14–175.98] ng/mL vs. 129.56 [98.51–172.95] ng/mL) compared to the systemic circulation (Appendix A).

### 2.3. Serum Levels of Parameters at Time 1

After treatment with mesoglycan, there was a statistically significant decrease in the serum levels of VCAM-1 (660.6 [629.3–743.7] ng/mL vs. 763.5 [685.7–849.1] ng/mL, *p* = 0.001), IL-6 (2.53 [1.28–3.52] pg/mL vs. 3.89 [1.95–5.20] pg/mL, *p* = 0.006), IL-8 (9.09 [6.86–13.90] pg/mL vs. 9.16 [6.86–14.76] pg/mL, *p* = 0.003), MMP-2 (46.89 [40.92–66.78] ng/mL vs. 51.48 [46.94–88.39] ng/mL, *p* = 0.0002), MMP-9 (36.22 [25.03–64.24] ng/mL vs. 69.02 [39.89–96.74] ng/mL, *p* = 0.003), SCD-1 (157.1 [140.5–179.4] ng/mL vs. 187.4 [167.2–235.7] ng/mL, *p* = 0.0005), and SDC-4 (42.22 [34.56–53.72] ng/mL vs. 56.75 [46.04–71.71] ng/mL, *p* = 0.0002) in the varicose veins at T1 compared to T0. There was a significant increase in TIMP-2 (172.40 [118.97–211.29] ng/mL vs. 127.29 [98.14–175.98] ng/mL, *p* = 0.003) in the varicose veins at T1 compared to T0. Similarly, non-significant changes were observed in ICAM-1 (25.95 [21.75–34.44] ng/mL vs. 23.72 [21.64–29.65] ng/mL, *p* = 0.89) and TGF-β (25.33 [21.46–34.40] ng/mL vs. 31.29 [20.55–46.15] ng/mL, *p* = 0.631) from varicose vein samples at T1 compared to T0 (Figure 1, Figure 2, Figure 3, Figure 4, Figure 5 and Figure 6).

The same parameters from systemic blood circulation showed that treatment with mesoglycan resulted in a statistically significant decrease in VCAM-1 (650.0 [587.5–727.0] ng/mL vs. 709.8 [627.4–826.0] ng/mL, *p* = 0.001), IL-6 (1.42 [0.97–3.18] pg/mL vs. 2.57 [1.67–3.77] pg/mL, *p* = 0.002), IL-8 (7.75 [4.32–13.41] pg/mL vs. 8.10 [4.91–17.96] pg/mL, *p* = 0.001), MMP-2 (42.84 [39.63–65.06] ng/mL vs. 43.57 [37.23–79.19] ng/mL, *p* = 0.0002), MMP-9 (35.03 [19.88–75.61] ng/mL vs. 52.91 [36.87–94.26] ng/mL, *p* = 0.0003), SCD-1 (148.5 [129.0–176.0] ng/mL vs. 177.2 [161.7–203.2] ng/mL, *p* = 0.0007), and SDC-4 (37.07 [33.41–49.26] ng/mL vs. 51.74 [38.27–62.31] ng/mL, *p* = 0.065) at T1 compared to T0. Levels of TIMP-2 (165.16 [122.59–197.94] ng/mL vs. 129.56 [98.51–172.95] ng/mL, *p* = 0.0002) increased significantly at T1 in antecubital veins, while non-significant changes were observed in ICAM-1 (24.13 [21.74–32.56] ng/mL vs. 23.18 [18.43–28.55] ng/mL, *p* = 0.38) and TGF-β (26.01 [23.52–32.98] ng/mL vs. 29.98 [21.98–44.32] ng/mL, *p* = 0.69) levels (Appendix A).

At the completion of the 12-week treatment with mesoglycan, significant changes in the parameters were observed both locally and systemically. Systemic serum levels of IL-6 (1.42 [0.97–3.18] pg/mL vs. 2.53 [1.28–3.52] pg/mL, *p* = 0.001), IL-8 (7.75 [4.32–13.41] pg/mL vs. 9.09 [6.86–13.90] pg/mL, *p* = 0.006), MMP-2 (42.84 [39.63–65.06] ng/mL vs. 46.89 [40.92–66.78] ng/mL, *p* = 0.03), SCD-1 (148.5 [129.0–176.0] ng/mL vs. 157.1 [140.5–179.4] ng/mL, *p* = 0.02), and SDC-4 (37.07 [33.41–49.26] ng/mL vs. 42.22 [34.56–53.72] ng/mL, *p* = 0.04) were significantly reduced at T1 when compared to values from varicose veins. Minor reductions were noted in VCAM-1 (650.0 [587.5–727.0] ng/mL vs. 660.6 [629.3–743.7] ng/mL, *p* = 0.138), and comparable values were found for ICAM-1 (24.13 [21.74–32.56] ng/mL vs. 25.95 [21.75–34.44] ng/mL, *p* = 0.78), MMP-9 (35.03 [19.88–75.61] ng/mL vs. 36.22 [25.03–64.24] ng/mL, *p* = 0.47), TGF-β (26.01 [23.52–32.98] ng/mL vs. 25.33 [21.46–34.40] ng/mL, *p* = 0.66), and TIMP-2 (165.16 [122.59–197.94] ng/mL vs. 172.40 [118.97–211.29] ng/mL, *p* = 0.27) between samples from systemic veins and varicose veins at T1 (Appendix A).

### 2.4. Signs and Symptoms at Time 0 and Time 1

At T0, 95.6% of patients reported signs and symptoms consistent with CVD, with a greater prevalence of heaviness in the legs (78.2%), edema (65.2%), cramps (56.5%), itching and pain (52.1%), sensation of heat (39.1%), and paresthesia (34.7%).

At T1, 69.5% of patients reported improvements in signs and symptoms compared to T0. Specifically, there was a statistically significant reduction in the following: heaviness in the legs (47.8%, *p* = 0.023), edema (34.7%, *p* = 0.023), cramps (13%, *p* = 0.004), itching (26%, *p* = 0.023), pain (30.4%, *p* = 0.041), sensation of heat (17.3%, *p* = 0.041), and paresthesia (8.6%, *p* = 0.023) (Table 2).

## 3. Discussion

Symptoms of MVC may manifest in the early stages of the disease, impacting CEAP classes C0 and C1. As the disease progresses and varicose veins develop (CEAP class C2), the signs and symptoms become more pronounced, with the associated pathophysiological changes showing increasing severity. Multiple studies have demonstrated the local increase in cytokines (IL-8, IL-6, and IL-12) [34,35], other inflammatory parameters (vWF), and endothelial damage (VCAM-1) [36] in varicose vein serums compared to what is evident systemically [37,38], and how these changes are partly reversible with phlebotonic therapy [36]. Our findings demonstrated elevated levels of VCAM-1, MMP-2, MMP-9, SCD-1, SDC-4, and IL-6 in serum obtained from varicose veins compared to that from the systemic circulation veins of the same subject. Compared to earlier studies in the literature, our findings offer a deeper understanding of the pathophysiology of CVD and the role of SDCs. Moreover, for the first time, it highlights the potential reversibility of endothelial and glycocalyx damage after a short treatment with a glycosaminoglycan-based drug.

Among the constituents of the cellular glycocalyx, SDCs are the molecules most commonly involved in inflammation, serving as reliable markers of damage [39,40]. In the presence of acute or chronic inflammation, the ectodomain expression of SDCs and their subsequent proteolytic cleavage by MMPs is greatly increased [41,42,43]. In our study, higher concentrations of SCD-1 (187.4 [167.2–235.7] ng/mL vs. 177.2 [161.7–203.2] ng/mL) and SDC-4 (56.75 [46.04–71.71] ng/mL vs. 51.74 [38.27–62.31] ng/mL) were found locally in subjects enrolled at baseline compared with systemic concentrations, demonstrating that MVC is a chronic progressive disease sustained by inflammation and damage to the endothelial glycocalyx. The effects of restoring the endothelial glycocalyx following mesoglycan administration were indirectly demonstrated by a reduction in SCD-1 (157.1 [140.5–179.4] ng/mL vs. 187.4 [167.2–235.7] ng/mL, *p* = 0.0005) and SDC-4 (42.22 [34.56–53.72] ng/mL vs. 56.75 [46.04–71.71] ng/mL, *p* = 0.0002) at the site of varicose veins, with similar findings observed in the systemic circulation (SCD-1: 148.5 [129.0–176.0] ng/mL vs. 177.2 [161.7–203.2] ng/mL, *p* = 0.0007; SDC-4: 37.07 [33.41–49.26] ng/mL vs. 51.74 [38.27–62.31] ng/mL, *p* = 0.065). Functional restoration of the glycocalyx is an important mechanism for preventing the development of cardiovascular diseases, including atherosclerosis, diabetes [42], and neurodegenerative diseases [43]. Studies have emphasized the close connection between the role of mesoglycan in promoting angiogenesis and SDC-4 [44]. Our findings are the first to suggest that mesoglycan influences SDC-1 and SDC-4 levels, although additional research is required to gain a deeper understanding of this interaction.

Chronic inflammation is implicated in the pathophysiology of several chronic diseases, such as atherosclerosis, tumor growth and metastasis progression, cellular senescence, and various degenerative processes [45]. In MVC, low-grade inflammation is sustained locally by leukocytes that, when activated, migrate and secrete several inflammatory markers [46]. Increased IL-6 values at the level of varicose veins are a sensitive marker of local inflammation associated with elevated intravascular pressure in the affected district. IL-6 levels are not only more sensitive in CVI [47] but also correlate significantly with age and disease severity [48], whereas IL-8 levels are more specific to CVI [49,50]. In our study, IL-6 and IL-8 values at T0 were significantly higher at the level of varicose veins compared to control veins (*p* = 0.0013 and *p* = 0.001, respectively), confirming the existence of an inflammatory microenvironment in the early stages of CVD. Although preliminary, our results not only demonstrate a significant local reduction of IL-6 and IL-8 but also highlight how the anti-inflammatory effect of mesoglycan extends systemically, effectively making it a potential therapeutic tool for counteracting the damage caused by chronic low-grade inflammation. Moreover, our data reveal a significant correlation between increased interleukin-6 and elevated SDC-1 levels, consistent with their known physiological relationship [51,52].

Elevated levels of MMP-2 and MMP-9, along with an altered MMP/TIMP ratio, contribute to the formation of varicose veins by perpetuating tissue damage across all layers of the venous wall, including the extracellular matrix (ECM) [52,53]. Some studies indicate that the distribution of MMPs varies across different layers of the vein wall, with their effects depending on the severity of CVD [52]. Our findings are consistent with the existing literature [53], demonstrating higher serum concentrations of MMP-2 and MMP-9 in samples from varicose veins compared to systemic circulation samples. Until now, a study conducted on diabetic patients with PAD [33] has shown a reduction in MMP-2 and MMP-9 levels following mesoglycan treatment. For the first time, our results highlight the local and systemic effects of mesoglycan in reducing MMP-2 and MMP-9 levels in patients with CVD classified as CEAP C2. The reduction of MMPs induced by mesoglycan in varicose veins contributes to decreased shedding of the endothelial glycocalyx, which in turn lowers circulating SDC levels. Therefore, our results support the use of mesoglycan in CVI as a modifying agent of the pathophysiological mechanisms involved in endothelial damage.

Moreover, our results demonstrate elevated VCAM-1 levels in blood samples derived from venous blood collected from the most peripheral region of a varicose vein compared to those from the systemic circulation, which can be attributed to elevated intravascular pressure in the venous district of the lower limbs, leading to EC activation. This study demonstrated the significant reduction of systemic and local VCAM-1 levels after mesoglycan treatment, suggesting that the beneficial impact on endothelial function is not limited to the vascular district subjected to increased intraluminal pressures but that normal endothelial function is restored systemically. This concept sheds new light on the importance of proper CVD therapy since its manifestations are systemic and closely related to cardiovascular disease, as evidenced by Prochaska’s findings. Serum ICAM-1 levels in varicose veins did not significantly differ from systemic values and were unaffected by mesoglycan treatment. We hypothesize that this outcome is primarily methodological, considering findings reported in similar studies [54,55,56]. ICAM-1 and VCAM-1 are endothelial proteins actively released through proteolysis in response to blood stasis damage. Studies have shown that after a short period of orthostatism (t = 30 min), their serum levels increase significantly (*p* = 0.0001 and *p* = 0.002, respectively). However, this increase is not observed under resting conditions (clinostatism) in either varicose veins or systemic circulation veins (*p* = 0.74 and *p* = 0.4, respectively) [54]. In our study, sampling was conducted with the patient in the orthostatic position, without specific consideration of the duration spent in orthostasis.

The findings concerning the serum levels of TGF-β in blood derived from varicose veins and the systemic circulation agree with the data collected in the literature documenting the imbalance in TGF-β production/activity in the pathogenesis of MVC. However, the results to date are inconclusive [57,58].

Our results are promising despite the limitations of our study, which are primarily the small sample size and the absence of a control group. Additionally, we acknowledge the lack of a standardized blood sampling procedure, including parameters such as exercise, time spent in orthostasis, and fasting periods. Variations in exercise, diet (time since the last meal), and the duration of orthostatism may have influenced the results of certain laboratory parameters, particularly chemokines. Compression therapy may also have played a similar role in affecting the outcomes. Our study underscores the importance of further investigating the molecular mechanisms underlying the restoration of the endothelial glycocalyx and endothelial dysfunction after the use of glycosaminoglycan-containing drugs, and this can be achieved with further in vivo and in vitro research. In addition, we consider that further monitoring of the parameters under investigation at intervals following the discontinuation of mesoglycan therapy could provide valuable insights into the stability of the results over time.

## 4. Materials and Methods

### 4.1. Study Design

This monocentric prospective interventional study was conducted at the Angiology and Non-invasive Vascular Diagnostics Service, Polyclinic Agostino Gemelli IRCCS (Rome, Italy).

The study protocol was approved by the ethics committee of the Catholic University of Rome and was conducted in accordance with the Declaration of Helsinki and its amendments. Suitable patients were enlisted by investigators and provided written informed consent to participate.

### 4.2. Patients

This study recruited consecutive subjects who spontaneously presented to our angiology unit between 1 January 2023 and 1 June 2024 for angiological examinations or ultrasonographic evaluation.

We included patients aged ≥ 18 of either sex with CVD of clinical stage 2 according to CEAP classification. Patients were excluded if they had a medical history of chronic inflammatory diseases, active infectious diseases or infections that were either systemic (such as bronchitis, pneumonia, or urinary tract infections) or localized to the lower limbs (lymphangitis, erysipelas, and skin ulcers) in the month preceding enrolment, diabetes mellitus, and cancer. We also excluded patients with a body mass index > 35 kg/m^2^, concomitant statin and phlebotonic drug therapy, corticosteroid therapy, concomitant non-steroidal anti-inflammatory drugs (NSAID) therapy or administered in the four weeks prior to enrollment, previously active thromboembolic venous disease (deep venous thrombosis, superficial venous thrombosis, and pulmonary embolism).

### 4.3. Assessments and Treatments

Patients fulfilling the inclusion and exclusion criteria were evaluated for somatic and demographic data (with regard to age, sex, familiarity for CVD, work activity, and pregnancy), along with a careful history of any comorbidities and therapies taken.

Data from the objective examination of the lower limbs were recorded, with particular attention to signs of venous disease (reticular varicose veins, varicosities, edema, dermatitis, eczema, white atrophy, and healed or active ulcers).

The evaluation of veins of the lower limbs was performed using high-resolution ultrasonography (Epiq Elite Philips Medical Systems, Monza, MB, Italy) and a linear 12-3 MHz transducer. The patient was placed in clinostatism and in orthostatism with the limb examined in discharge and integrated with hemodynamic tests (i.e., Valsalva maneuvers and the dynamic squeezing test). For the deep venous system, parameters such as patency and the presence or absence of reflux (with specific measurements of reflux extension and duration) were assessed. For the superficial venous system, the same parameters were evaluated in the saphenous veins (great and small saphenous veins) and other relevant venous vessels (Giacomini vein, Leonardo vein, and major collaterals). For the perforator system, the presence and anatomical localization of any incompetent perforating veins were described.

At baseline, we collected blood sampling with an enzyme-linked immunosorbent assay (ELISA) tube (R&D Systems, Minneapolis, MN, USA) from the antecubital vein and from the most distal varicose vein of the lower legs to evaluate serum cytokines and some markers of endothelial damage, particularly sICAM-1, sVCAM-1, VEGF, IL-6, IL-8, TGF-beta, MMP-2, syndecan-1, and syndecan-4. The patients were then treated with phlebotonic drugs as standard clinical practice, specifically mesoglycan at a dose of 50 mg every 12 h. This was administered orally for a period of 90 days. After 12 weeks, blood tests were repeated from the same sites as baseline to assess the same parameters. Any adverse effects related to the blood sampling site or the pharmacological treatment were also recorded.

Moreover, patients were questioned whether they wore graduated compression elastic stockings (GCS) during the period of treatment. If patients were wearing GCS, as per previous medical indications, data were recorded. Data on compressive treatment compliance were recorded.

### 4.4. Statistical Analysis

Demographic and clinical characteristics, including age, sex, body mass index (BMI), and comorbidities, were summarized as means and percentages, with categorical variables reported as counts and percentages. Changes in biomarker levels from samples obtained from both varicose veins and the systemic circulation, as well as reductions in patient-reported symptoms, were analyzed using the paired Wilcoxon test to compare values between T0 and T1. The frequency of adverse events related to mesoglycan therapy and patient compliance were similarly assessed with the paired Wilcoxon test. Additionally, comparisons of samples from the two anatomical districts at T0 and T1 were performed using the same statistical method for non-paired samples. The analysis was conducted using R version 3.6.2. *p*-values < 0.05 were considered statistically significant.

## 5. Conclusions

Our study, although conducted on a small number of patients and over a short period, aimed to investigate various markers believed to indicate vascular endothelial damage in the early stages of MVC (CEAP C2). Blood samples were drawn simultaneously from an arm and a leg in the orthostatic position from the same patient. Differences in inflammatory and endothelial damage markers between the systemic circulation and varicose veins were observed in subjects in the early stages of CVI. Despite the limitations, mesoglycan therapy at a dose of 100 mg/day administered for 3 months demonstrates an improvement in both local and systemic inflammation. Systemic treatment with mesoglycan resulted, indirectly, in the restoration of the integrity of the endothelial glycocalyx. Mesoglycan could be used in the early stages of CVI as a drug capable of arresting or slowing down some of the underlying pathophysiological mechanisms. In the context of personalized precision medicine and given the growing scientific interest in syndecans, this study provides new insights for potential treatments in chronic venous disease and beyond.

## Figures and Tables

**Figure 1 ijms-26-04046-f001:**
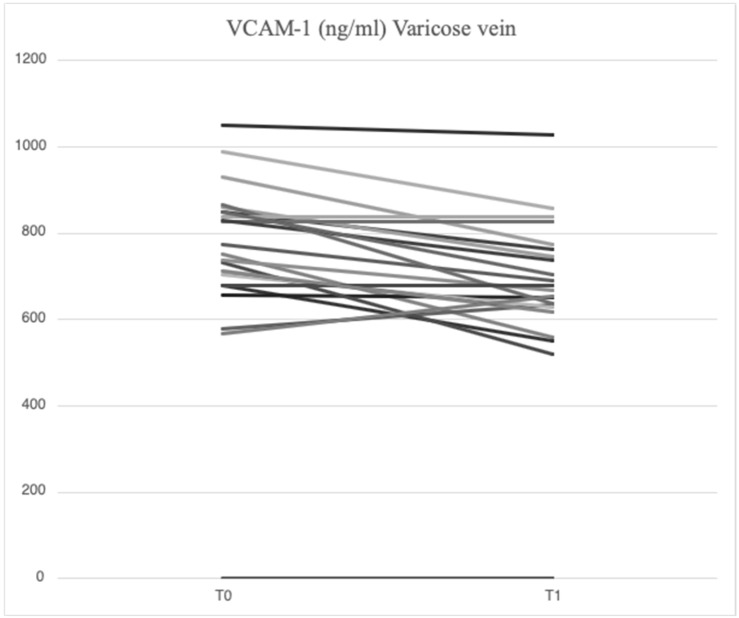
Linear graph representing the effects of mesoglycan on VCAM-1 (ng/mL) levels from blood derived from varicose veins at baseline (T0) and after 90 days of treatment (T1). Each line of the multiple-line chart represents each of the 23 patients involved in this study. A statistically significant decrease in VCAM-1 values can be observed (660.6 [629.3–743.7] ng/mL at T1 vs. 763.5 [685.7–849.1] ng/mL at T0; *p* = 0.001). Statistical analysis was performed using a paired Wilcoxon test and *p* < 0.05 was considered statistically significant.

**Figure 2 ijms-26-04046-f002:**
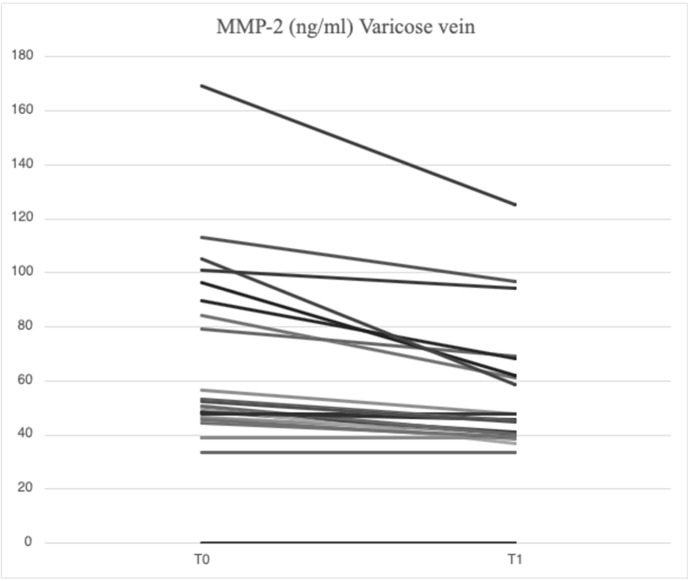
Linear graph representing the effects of mesoglycan on MMP-2 (ng/mL) levels from blood derived from varicose veins at baseline (T0) and after 90 days of treatment (T1). Each line of the multiple-line chart represents each of the 23 patients involved in this study. A statistically significant decrease in MMP-2 values can be observed (46.89 [40.92–66.78] ng/mL at T1 vs. 51.48 [46.94–88.39] ng/mL at T0; *p* = 0.0002). Statistical analysis was performed using a paired Wilcoxon test and *p* < 0.05 was considered statistically significant.

**Figure 3 ijms-26-04046-f003:**
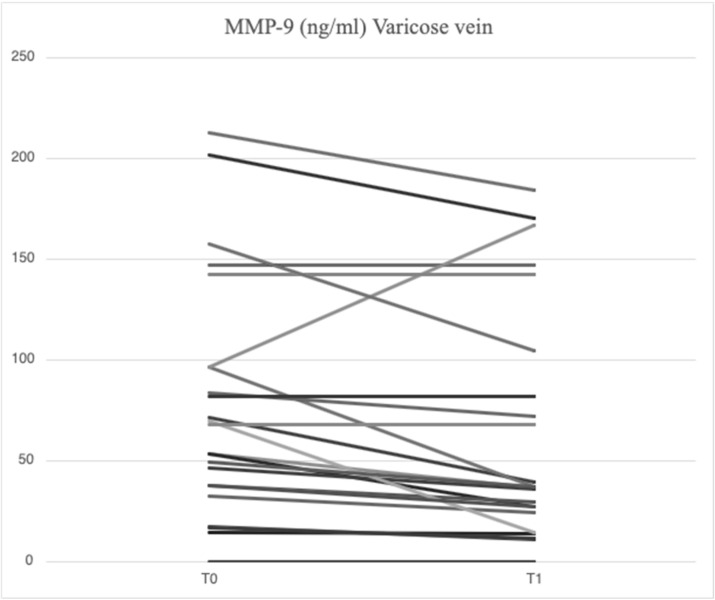
Linear graph representing the effects of mesoglycan on MMP-9 (ng/mL) levels from blood derived from varicose veins at baseline (T0) and after 90 days of treatment (T1). Each line of the multiple-line chart represents each of the 23 patients involved in this study. A statistically significant decrease in MMP-9 values can be observed (36.22 [25.03–64.24] ng/mL at T1 vs. 69.02 [39.89–96.74] ng/mL at T0; *p* = 0.003). Statistical analysis was performed using a paired Wilcoxon test and *p* < 0.05 was considered statistically significant.

**Figure 4 ijms-26-04046-f004:**
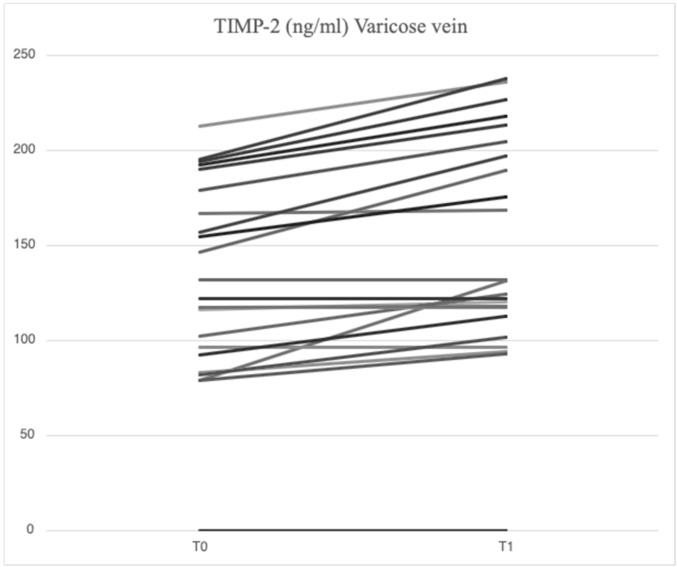
Linear graph representing the effects of mesoglycan on TIMP-2 (ng/mL) levels from blood derived from varicose veins at baseline (T0) and after 90 days of treatment (T1). Each line of the multiple-line chart represents each of the 23 patients involved in this study. A statistically significant increase in TIMP-2 values can be observed (172.40 [118.97–211.29] ng/mL at T1 vs. 127.29 [98.14–175.98] ng/mL at T0; *p* = 0.003). Statistical analysis was performed using a paired Wilcoxon test and *p* < 0.05 was considered statistically significant.

**Figure 5 ijms-26-04046-f005:**
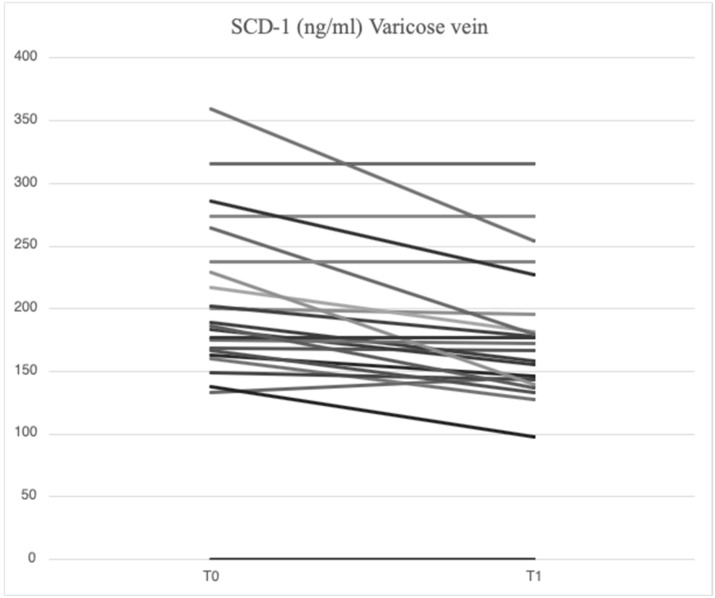
Linear graph representing the effects of mesoglycan on SDC-1 (ng/mL) levels from blood derived from varicose veins at baseline (T0) and after 90 days of treatment (T1). Each line of the multiple-line chart represents each of the 23 patients involved in this study. A statistically significant decrease in SDC-1 values can be observed (157.1 [140.5–179.4] ng/mL at T1 vs. 187.4 [167.2–235.7] ng/mL at T0; *p* = 0.0005). Statistical analysis was performed using a paired Wilcoxon test and *p* < 0.05 was considered statistically significant.

**Figure 6 ijms-26-04046-f006:**
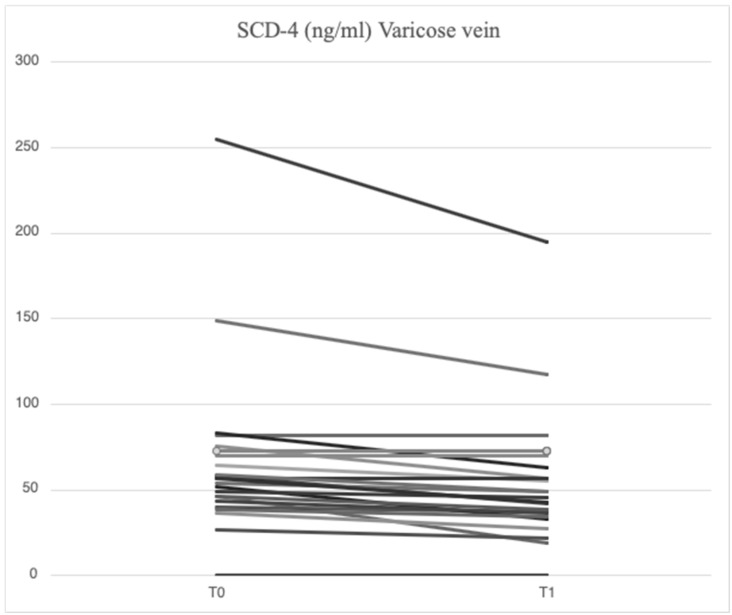
Linear graph representing the effects of mesoglycan on SDC-4 (ng/mL) levels from blood derived from varicose veins at baseline (T0) and after 90 days of treatment (T1). Each line of the multiple-line chart represents each of the 23 patients involved in this study. A statistically significant decrease in SDC-4 values can be observed (42.22 [34.56–53.72] ng/mL at T1 vs. 56.75 [46.04–71.71] ng/mL at T0; *p* = 0.0002). Statistical analysis was performed using a paired Wilcoxon test and *p* < 0.05 was considered statistically significant.

**Table 1 ijms-26-04046-t001:** Table summarizing the characteristics of patients.

Characteristics of the Patients	
No.	23
Median age (IQR)—yr	60 (23–74)
Female sex—No. (%)	18 (78.2%)
Caucasian—No. (%)	22 (95.6%)
Median height (IQR)—cm	165 (158–192)
Median weight (IQR)—kg	68 (52–102)
Body mass index > 25 kg/m^2^—No. (%)	9 (39.1%)
Inheritance—No. (%)	19 (82.6%)
Pregnancies—No. (%)	13 (72.2%)
Sedentary lifestyle—No. (%)	15 (65.2%)
Smoking habit—No. (%)	6 (26%)

**Table 2 ijms-26-04046-t002:** Comparison of signs and symptoms at T0 and T1.

	Population (No. = 23)		
	T0	T1	*P*
Heaviness—No. (%)	18 (78.2%)	11 (47.8%)	0.023
Edema—No. (%)	15 (65.2%)	8 (34.7%)	0.023
Cramps—No. (%)	13 (56.5%)	3 (13%)	0.004
Itching—No. (%)	12 (52.1%)	6 (26%)	0.023
Pain—No. (%)	12 (52.1%)	7 (30.4%)	0.041
Heat—No. (%)	9 (39.1%)	4 (17.3%)	0.041
Paresthesia—No. (%)	8 (34.7%)	2 (8.6%)	0.023

## Data Availability

Data is contained within the article and Appendix A.

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
