# Peer review of "Local and Systemic Endothelial Damage in Patients with CEAP C2 Chronic Venous Insufficiency: Role of Mesoglycan"

_ijms, 2025, doi:10.3390/ijms26094046_

Round 1

Reviewer 1 Report

Comments and Suggestions for Authors

Dear Authors,

I find this paper very interesting, while this is an up-to-date topic.

I have several comments and suggestions in order to improve this paper:

  • If I'm not mistaken, the last sentence in the abstract is written in a smaller font size than the rest of the text.
  • Introduction, second paragraph - I do not consider that so many details regarding about CEAP classification are necessary.
  • The introduction section does not cover the current knowledge regarding other blood markers in CVD patients. One paragraph discussing also the role of circulating markers in primary chronic venous insufficiency should be introduced (https://pubmed.ncbi.nlm.nih.gov/26650463/, https://pubmed.ncbi.nlm.nih.gov/35978924/ , https://pubmed.ncbi.nlm.nih.gov/23928282/).
  • Introduction section: „Pharmacotherapy of CVD is used in all stages of the disease, both to counteract the physiopathological mechanisms that support tissue damage, and to improve the severity of symptoms, the quality of life and prevent the most unfortunate consequences of the disease. In this context, a wide range of venoactive drugs are recognised...”. First, I consider that this paragraph should be moved in the discussion section. Also, considering that you mentioned that a „wide range of venoactive drugs” are avialble, you should mention the most improtant pharmacological active substances (https://www.mdpi.com/2227-9059/10/5/1076, https://www.mdpi.com/2813-2475/3/1/4, https://pubmed.ncbi.nlm.nih.gov/28225220/, https://pubmed.ncbi.nlm.nih.gov/28124877/, https://pubmed.ncbi.nlm.nih.gov/33141449/), not only those you mentioned.
  • Methods section, 2.2. Patients: You excluded patients with concomitant statin and phlebotonic drug therapy. I agree, but you should support this exclusion criteria with literature data (https://pubmed.ncbi.nlm.nih.gov/34877912/, https://pubmed.ncbi.nlm.nih.gov/35373942/). Several data regarding how this drugs act on the venous wall in the introduction section would make more valid these exclusion criterion. A clear presented background would support the aim of this study and your hipotesis.
  • Discussion section, penultimate paragraph: „In addition, we believe that further monitoring of the parameters...”. I recommend to change „believe” with „consider”.
  • Some further research directions may be mentioned.

Hoping that my suggestions and the recommended referecnes will be helpful for you, I wish you good luck in publishing this paper!

Author Response

Comment 1: If I'm not mistaken, the last sentence in the abstract is written in a smaller font size than the rest of the text.

Response 1: Thank you for highlighting this issue. We have revised the sentence under consideration.

Comment 2: Introduction, second paragraph - I do not consider that so many details regarding about CEAP classification are necessary.

Response 2: Thank you for highlighting this aspect of the text. We agree that this paragraph can be more concise, so the text has been modified.

Comment 3: The introduction section does not cover the current knowledge regarding other blood markers in CVD patients. One paragraph discussing also the role of circulating markers in primary chronic venous insufficiency should be introduced (https://pubmed.ncbi.nlm.nih.gov/26650463/, https://pubmed.ncbi.nlm.nih.gov/35978924/ , https://pubmed.ncbi.nlm.nih.gov/23928282/).

Response 3: We agree that emphasizing this topic may give more importance to our results. Therefore, a short paragraph has been added in the text as requested.

Comment 4: Introduction section: „Pharmacotherapy of CVD is used in all stages of the disease, both to counteract the physiopathological mechanisms that support tissue damage, and to improve the severity of symptoms, the quality of life and prevent the most unfortunate consequences of the disease. In this context, a wide range of venoactive drugs are recognised...”. First, I consider that this paragraph should be moved in the discussion section. Also, considering that you mentioned that a „wide range of venoactive drugs” are avialble, you should mention the most improtant pharmacological active substances (https://www.mdpi.com/2227-9059/10/5/1076, https://www.mdpi.com/2813-2475/3/1/4, https://pubmed.ncbi.nlm.nih.gov/28225220/, https://pubmed.ncbi.nlm.nih.gov/28124877/, https://pubmed.ncbi.nlm.nih.gov/33141449/), not only those you mentioned.

Response 4: Thank you for your comment. We are sorry to tell you that this paragraph has not been moved into the discussion because we believe it is critical in this context for the introduction of the topic. A brief mention of the most relevant phlebotonic drugs has been included.

Comment 5: Methods section, 2.2. Patients: You excluded patients with concomitant statin and phlebotonic drug therapy. I agree, but you should support this exclusion criteria with literature data (https://pubmed.ncbi.nlm.nih.gov/34877912/, https://pubmed.ncbi.nlm.nih.gov/35373942/). Several data regarding how this drugs act on the venous wall in the introduction section would make more valid these exclusion criterion. A clear presented background would support the aim of this study and your hipotesis.

Response 5: We fully agree to include a brief background on the role of statins in CVD. The text has been edited and appropriate references have been included.

Comment 6: Discussion section, penultimate paragraph: „In addition, we believe that further monitoring of the parameters...”. I recommend to change „believe” with „consider”.

Response 6: Thank you for your comment.

Reviewer 2 Report

Comments and Suggestions for Authors
  1. The manuscript is more descriptive without mechanistic insights. Current data could not support the conclusion of this study.
  2. The causal relationship between mesoglycan and endothelial damage in patients with CEAP should be established. The in vitro or in vivo models are highly recommended. 
  3. Many scientific descriptions do not have proper reference citations, so please check them throughout the manuscript.
  4. The style of references should be revised according to the author’s guidelines.

Author Response

Comment 1: The manuscript is more descriptive without mechanistic insights. Current data could not support the conclusion of this study.

Response 1: thank you for your response, we agree that we do not provide much mechanistic data, but that was not the purpose of our work. However, our data are in line with studies that have already been published in the literature, including recent ones, regarding the use of glycosaminoglycan-containing drugs (doi: 10.3389/fimmu.2023.1172892. - doi: 10.3390/ijms18030572). In addition, our study is the first to provide data regarding the effects of mesoglycan on syndecans, components of the cellular glycocalyx, in CVD patients.

Comment 2: The causal relationship between mesoglycan and endothelial damage in patients with CEAP should be established. The in vitro or in vivo models are highly recommended.

Response 2: Thank you for your response, a paragraph was inserted on page 14 - the last paragraph of the discussion. The purpose of our study was to evaluate in symptomatic and non-symptomatic CEAP C2 CVD subjects the systemic and local effects of cellular glycocalyx damage and endothelial dysfunction. No molecular investigations or further mechanistic insights were planned, but certainly our results will open this new research direction.

Comment 3: Many scientific descriptions do not have proper reference citations, so please check them throughout the manuscript.

Response 3: Thank you for your comment. We made this clarification.

Comment 4: The style of references should be revised according to the author’s guidelines.

Response 4: Thank you for raising this issue, the text has been corrected.

Reviewer 3 Report

Comments and Suggestions for Authors

SUMMARY

     The authors of this manuscript at Fondazione Policlinico University, Catholic University of the Sacred Heart and University of Ferrara-, in Italy, report outcomes of a prospective, monocentric study of aspects of the pathophysiology of chronic venous disease (CVD) involving inflammation-associated structural alterations of the glycocalyx, the glycolipid and glycoprotein coating of the apical surface of endothelial cells facing the lumen of blood vessels, which contribute to regulation of recruitment and migration of immune cells from the blood into tissues of the body.   Authors of this research focus on inflammatory damage to glycocalyx coatings and describe how treatment with mesoglycan, a glycosaminoglycan (GAG) derived from animal lung and intestinal tissue, composed primarily of heparan and dermatan sulfate, may contribute to improved prognoses of CVD patients.    The investigation was based on data from 23 CVD patients with CEAP C2 classification of the severity of varicose veins, which includes asymptomatic patients (C2A); and symptomatic (C2S), along with measurement of inflammatory biomarkers in the peripheral blood serum.    Outcomes of this study demonstrate that mesoglycan-treated patients demonstrated diminished levels of inflammatory biomarkers, thus indicating improved CVD prognoses correlating with and probably resulting from administration of this GAG.  This report underscores the future promise for use of mesoglycan, especially since it seems well tolerated by the test subjects.   The topic material of this report is timely and addresses a major medical need.  It is anticipated that it will stimulate reader interest and also serve as a motivator for expanded investigation how mesoglycan and other GAGs may be incorporated into emerging countermeasures for CVD and other vascular disorders with  pathogeneses related to disrupted glycocalyx function.

.      This is a clearly written and very useful paper. This reviewer here conditionally recommends publication if the authors are willing to make revisions in the present manuscript according to the comments below:

REVIEWER COMENTS

Comment 1.  In the Abstract. The authors should state that mesoglycan, is a glycosaminoglycan (GAG)

Comment 2.  In the Abstract, the full length terms for each abbreviation should precede it, along with a definition of CEAP C2, which in the interests of staying within the word limit for the Abstract might be condensed into a single sentence stated above in the Summary for this review, specifically:  ”… CEAP C2 classification of the severity of varicose veins, which includes asymptomatic patients (C2A); and symptomatic (C2S),…”, or words to that effect.

Comment 3.  Again, taking into consideration the journal`s word limit for the Abstract, the authors should include route of administration and time frame for treatment with mesoglycan.

Comment 4.  In Materials and Methods on page 5, first use of the abbreviation ELISA should be in parentheses and preceded by the full length term: enzyme-linked immunoassay.

In Materials and Methods, Section 2.2, the phrase “….The study provided the enrolment of consecutive subjects spontaneously afferent to…” may not be  completely clear to readers and should be rephrased, perhaps using words other than “spontaneously afferent”.  As stated it is a bit confusing.  Suggestion:  Try and use the word “recruited” in the sentence.

Author Response

Comment 1: In the Abstract. The authors should state that mesoglycan, is a glycosaminoglycan (GAG).

Response 1: Thank you for your comment. We made this clarification.

Comment 2: In the Abstract, the full length terms for each abbreviation should precede it, along with a definition of CEAP C2, which in the interests of staying within the word limit for the Abstract might be condensed into a single sentence stated above in the Summary for this review, specifically:  ”… CEAP C2 classification of the severity of varicose veins, which includes asymptomatic patients (C2A); and symptomatic (C2S),…”, or words to that effect.

Response 2: Thank you for raising this issue, the text has been corrected.

Comment 3: Again, taking into consideration the journal`s word limit for the Abstract, the authors should include route of administration and time frame for treatment with mesoglycan.

Response 3: Thank you for your comment. We made this clarification.

Comment 4: In Materials and Methods on page 5, first use of the abbreviation ELISA should be in parentheses and preceded by the full length term: enzyme-linked immunoassay.

Response 4: Thank you for raising this issue, the text has been corrected.

Comment 5: In Materials and Methods, Section 2.2, the phrase “….The study provided the enrolment of consecutive subjects spontaneously afferent to…” may not be  completely clear to readers and should be rephrased, perhaps using words other than “spontaneously afferent”.  As stated it is a bit confusing.  Suggestion:  Try and use the word “recruited” in the sentence.

Response 5: Thank you for raising this issue, the text has been corrected.

Round 2

Reviewer 1 Report

Comments and Suggestions for Authors

Dear Authors,

I consider that the revised version of your paper presents the details in a clear manner, and is improved from the scientific rigour point of view.

I enjoyed reading this article, and I hope you will get it published soon!

Kind regards,

Reviewer.

Reviewer 2 Report

Comments and Suggestions for Authors

The authors have addressed my comments.